# Comprehensive Expression Analysis of the WRKY Gene Family in *Phoebe bournei* under Drought and Waterlogging Stresses

**DOI:** 10.3390/ijms25137280

**Published:** 2024-07-02

**Authors:** Zhongxuan Wang, Limei You, Na Gong, Can Li, Zhuoqun Li, Jun Shen, Lulu Wan, Kaijin Luo, Xiaoqing Su, Lizhen Feng, Shipin Chen, Wenjun Lin

**Affiliations:** College of Forestry, Fujian Agriculture and Forestry University, Fuzhou 350002, China; wang_zx_1998@126.com (Z.W.); yolimy_y@126.com (L.Y.); gn2015477112@163.com (N.G.); lcknight1999@163.com (C.L.); 13285946066@163.com (Z.L.); zzshenjunjun@126.com (J.S.); wan201920@163.com (L.W.); luo10052021@126.com (K.L.); fjsxqsxq@126.com (X.S.); fjflz@126.com (L.F.)

**Keywords:** *Phoebe bournei*, Lauraceae, WRKY, drought stress, waterlogging stress

## Abstract

In response to biotic and abiotic stresses, the WRKY gene family plays a crucial role in plant growth and development. This study focused on *Phoebe bournei* and involved genome-wide identification of WRKY gene family members, clarification of their molecular evolutionary characteristics, and comprehensive mapping of their expression profiles under diverse abiotic stress conditions. A total of 60 WRKY gene family members were identified, and their phylogenetic classification revealed three distinct groups. A conserved motif analysis underscored the significant conservation of motif 1 and motif 2 among the majority of *PbWRKY* proteins, with proteins within the same class sharing analogous gene structures. Furthermore, an examination of cis-acting elements and protein interaction networks revealed several genes implicated in abiotic stress responses in *P. bournei*. Transcriptomic data were utilized to analyze the expression patterns of WRKY family members under drought and waterlogged conditions, with subsequent validation by quantitative real-time PCR (RT-qPCR) experiments. Notably, *PbWRKY55* exhibited significant expression modulation under drought stress; *PbWRKY36* responded prominently to waterlogging stress; and *PbWRKY18*, *PbWRKY38*, and *PbWRKY57* demonstrated altered expression under both drought and waterlogging stresses. This study revealed the *PbWRKY* candidate genes that potentially play a pivotal role in enhancing abiotic stress resilience in *P. bournei*. The findings have provided valuable insights and knowledge that can guide further research aimed at understanding and addressing the impacts of abiotic stress within this species.

## 1. Introduction

Abiotic stress factors such as drought and waterlogging seriously affect the growth and development process of trees and seriously limit the production and sustainable development of forestry. Drought is one of the important abiotic stress factors that hinder plant growth and development, and it is also one of the most common forms of plant damage. Waterlogging stress results in root hypoxia, which induces a series of physiological changes. Unfortunately, prolonged anaerobic respiration can result in the accumulation of harmful substances in the rhizosphere. With global warming and environmental degradation, these problems are becoming more and more significant. Therefore, improving the drought resistance and waterlogging resistance of forest trees has become the focus of current research on forest tree breeding.

*Phoebe bournei* (Hemsl.) Yang, an evergreen broad-leaved tree species classified under the genus *Phoebe* in the Lauraceae family, holds the status of a class II protected plant in China because of its profound ecological, medicinal, and economic value. Renowned for its fine wood with a beautiful texture, robust corrosion resistance, and minimal susceptibility to cracking during drying, this tree species has been widely used in shipbuilding, construction, and wood artistry [1]. However, *P. bournei* faces challenges such as drought, waterlogging, and salinization, leading to a decline in population size and scattered distribution patterns [2]. Despite its environmental relevance, the intricate response mechanism of *P. bournei* to various stressors remains relatively unexplored.

The WRKY gene family is an important transcriptional regulator that plays an important regulatory role in plant resistance to biotic and abiotic stresses. The structural domain of WRKYGQK consists of approximately sixty strongly conserved amino acid residues, with an invariant heptapeptide amino acid sequence at the end of the N-terminus. There are two zinc-finger motifs at the C-terminus, C_2_H_2_ (C-X_4–5_-C-X_22–23_-H-X-H) or C_2_HC (C-X_7_-C-X_23_-H-X-C) [3]. Studies have shown that an amino acid in WRKYGQK is substituted, resulting in a change in its sequence, including WRKYGKK, WRKYGEK, WRKYGRK, and WRKYGQK. Additionally, the amino acids RR, SK, KR, VK, IK, RM, RI, or KK of the WRKY structural domain often replace the original “RK” [4]. The WRKY transcription factor (TF) can specifically bind to the W-box (TTGACC/T) in the promoter of target genes to activate or repress transcription to regulate the expression of downstream genes [5]. The WRKY family is classified according to the number of WRKY structural domains and the type of zinc-finger motif. Group I contains two WRKY conserved domains, and the zinc-finger structure is C_2_H_2_. Groups II and III both contain only one WRKY conserved domain. The Group II zinc-finger structure is the same as that of Group I, which is C_2_H_2_, while the Group III zinc-finger structure is C_2_HC. Additionally, according to conserved structural motifs other than WRKYGQK, Group II can be divided into IIa, IIb, IIc, IId, and IIe [6,7].

Many studies have reported that WRKY TFs regulate gene expression and play a variety of biological functions, especially in abiotic stress responses [8,9]. The overexpression of an *Arabidopsis thaliana* WRKY TF (*AtWRKY30*) significantly enhances the ability of wheat to resist heat and drought [10], and *AtWRKY53* negatively regulates drought resistance in *A. thaliana* by mediating stomatal movement [11]. *AtWRKY57* can improve *A. thaliana* drought resistance by increasing the level of abscisic acid ABA [12]. In rice, *OsWRKY11* plays a significant role in rice resistance to high temperature and drought, improving the stress resistance of plants [13]. Similarly, the expression of *OsWRKY45* increases significantly under ABA, NaCl, and polyethylene glycol PEG treatments, and the drought resistance of *A. thaliana* is enhanced [14]. *Phyllostachys edulis*, *PheWRKY72-2* can enhance drought tolerance in transgenic *A. thaliana* by regulating stomatal closure [15]. In *Camellia japonica*, *CjWRKY21* is overexpressed during low-temperature pressure, and *CjWRKY22* is overexpressed during drought damage [16].

Although the WRKY gene family has been thoroughly researched in many plants, such as *A. thaliana* [17], *Zea mays* [18], *Oryza sativa* [13], and other plants, there has been no systematic analysis of the WRKY gene family in *P. bournei*. In particular, little is known about the response of *P. bournei* to abiotic stresses at the molecular level. In this study, members of the WRKY gene family were identified, and the phylogenetic relationship, gene structure, motif composition, cis-acting elements, exon–intron structure, evolutionary pattern, protein structure and interaction regulatory network, and gene expression patterns under waterlogging and drought stress were systematically analyzed. Additionally, the expression level of *PbWRKY* with different expression patterns was verified by a quantitative real-time PCR (RT-qPCR) analysis. This study provides a scientific basis for the in-depth study of the biological function of *PbWRKY* genes and their response mechanism in abiotic stress and also provides theoretical guidance for breeding *P. bournei* varieties with excellent resistance.

## 2. Results

### 2.1. The WRKY Gene Family in P. bournei

The conserved domain of the *P. bournei* WRKY was determined by the *A. thaliana* WRKY model. Finally, 60 WRKY were obtained and named *PbWRKY1*–*PbWRKY60* (Table 1). The physical and chemical properties (including amino acids, molecular weight, isoelectric point, and subcellular location) of the WRKY protein were analyzed by ExPaSy. Among the 60 WRKY proteins, there was a significant difference in the number of amino acids, with a minimum number of amino acids (104) and a maximum number of amino acids (883), a molecular weight range of 12.64 to 96.25 kDa, and an isoelectric point range of 4.80 to 10.80. Subcellular localization predicted that many *PbWRKY* proteins belong to nuclear proteins, while only *PbWRKY52* may belong to chloroplasts or nuclei.

### 2.2. PbWRKY Protein Phylogenetic Analysis

There were three groups of *PbWRKY* proteins (Figure 1): Group I in yellow, which contains 13 *PbWRKYs*; Group IIa in red, which contains 3 *PbWRKYs*; Group IIb in green, which contains five *PbWRKYs*; Group IIc in blue, which contains 16 *PbWRKYs*; Group IId in orange, which contains five *PbWRKYs*; Group IIe in purple, which contains 11 *PbWRKYs*; and the number of members was lowest in Group IIa and highest in Group IIc. The proportion of *PbWRKY* genes in each group was different from that of the *AtWRKY* genes in *A. thaliana*. Group I had the highest number of WRKY proteins in *A. thaliana*, but Group II had the most predominant WRKY proteins in *P. bournei*.

### 2.3. PbWRKY Multiple Sequence Alignment

*PbWRKY* domains and conserved motifs spanning 60 amino acids were investigated using multiple sequence alignment (Figure 2). The investigation of the *PbWRKY* subfamily conserved domains revealed that the members of the subfamily contained highly conserved WRKY domains, individual members with *WRKY*GQK heptapeptide domain variants, and zinc-finger structure variants. Group I contained two WRKY conserved domains. They were classified as I-N and I-C according to their position in the sequence. Group I-N *PbWRKYs* contained a *WRKY*GQK domain and a CX4CX22HXH zinc-finger structure. Group I-C *PbWRKYs* contained a *WRKY*GQK domain and a CX4CX23HXH zinc-finger structure. Relative to Groups I, II, and III, only one conserved domain exists. Group II was further divided into five distinct groups (Group IIa–Group IIe) based on the amino acid sequence differences. Groups IIa, IIb, IId, and IIe were similar, containing a *WRKY*GQK domain and a CX5CX23HXH zinc-finger structure. A *WRKY*GQK domain and a CX4CX23HXH zinc-finger structure were found in Group IIc *PbWRKYs*. A *WRKY*GQK domain and a CX7CX23HXC zinc-finger structure were found in Group III *PbWRKYs*.

### 2.4. PbWRKY Gene Structure and Motif Composition

Sixty *PbWRKY* protein sequences contained 20 motifs: the number of motifs in *PbWRKYs* ranged from one (*PbWRKY1*) to nine (*PbWRKY17*, *PbWRKY24*, *PbWRKY28*, and *PbWRKY57*) (Figure 3B). Motifs 1, 2, and 20 were observed in Group III of *PbWRKY* proteins, but motifs 1, 2, 4, 8, 11, 16, and 18 were identified in Group IId, whereas motifs 1, 2, 10, and 12 were observed in Group IIa. *PbWRKYs* all had motifs 1 and 2, while motifs 3, 9, 13, and 17 were only identified in Group I, and motifs 5, 6, and 18 were only identified in Group IId. Motifs 15 and 19 were unique to Groups IIc and IIe. Most branches containing motifs 4 and 16 were present in Group I, but *PbWRKY10*, which contained motif 16, was located in Group IIc.

The exon/intron structure and number of exons were analyzed to detect features of the evolutionary events in the *PbWRKY* genes family in *P. bournei* (Figure 3C). The *PbWRKY* gene contained two to eight exons, with *PbWRKY1* and *PbWRKY52* having eight exons (*n* = 8). *PbWRKYs* had similar exons and introns in the same group, such as in Groups IIa and IId, which had a low number of exons and a conserved structure. *PbWRKY* Group III contained two to three exons, with approximately 86% (6/7) containing three exons. Group IIe contained three to seven exons, with approximately 64% (7/11) containing three exons, and Group IIb contained four to seven exons, with approximately 60% (3/5) containing five exons. In conclusion, almost all *PbWRKYs* of the same group or subgroup had very similar motif compositions and gene structures. This indicated that these *PbWRKY* proteins may have similar functions, further validating the phylogenetic relationship of *PbWRKYs*.

### 2.5. Ka and Ks of the PbWRKY Gene Family

The origin and evolution of the *PbWRKYs* were further identified by calculating the *K*a and *K*s values for homologous gene pairs. We identified a total of 21 homologous gene pairs in the *P. bournei* genome (Table 2). The *K*s values ranged from 0.07 to 3.14 for *P. bournei* homologous pairs, indicating that the duplication event occurred approximately 5.4–241.94 million years ago, whereas only six homologous pairs were identified in *P. bournei* and *A. thaliana* (Table 3). The *K*s values for *P. bournei* and *A. thaliana* homologous gene pairs ranged from 1.82 to 4.03, suggesting that the replication event occurred between 140.37 and 309.77 million years ago. To understand the evolutionary selection pressure on different *PbWRKY* gene pairs, we also calculated the *K*a/*K*s ratio. *K*a/*K*s indicates a positive or diversifying selection when *K*a or *K*s is >1 and a neutral selection when *K*a or *K*s = 1. The results showed that the *K*a/*K*s ratios of the 21 *PbWRKY* homologous gene pairs were all <0.5 (Table 2), indicating that the WRKY gene family of *P. bournei* had undergone strong purifying selection during evolution, eliminating most of the deleterious mutations and resulting in a high level of protein conservation. Six *P. bournei* and *A. thaliana* homologous gene pairs (except *PbWRKY37-AtWRKY34*) had *K*a/*K*s ratios that were all <0.4 (Table 3), indicating that the WRKY gene families of *P. bournei* and *A. thaliana* have undergone mainly strong negative selection during evolution.

### 2.6. Analysis of Cis-Acting Elements in the Promoter Region of PbWRKYs

The cis-acting elements of the promoter sequence of *PbWRKYs* were obtained using the PlantCARE database to understand and speculate their possible response mechanisms under various stresses. The 56 cis-elements identified could be classified into four categories according to their functions, including 26 light-responsive elements, 12 hormone-related elements, 12 development elements, and six abiotic stress-responsive elements (Figure 4). A total of 111 G-boxes were the most abundant among all cis-elements, in addition to 48 Box 4 elements and 39 GT1 motifs, suggesting that the expression of *PbWRKYs* was mainly induced by light. Additionally, 100 ABA response elements (ABREs), 52 CGTCA motifs, and 27 TCA elements (salicylic acid response elements) were identified, indicating that *PbWRKYs* play a large part in the hormone response process. Significantly, many stress response elements were identified in the *PbWRKYs*, including 84 ARE elements (anaerobic elements), thirty MBS elements (drought response elements), and sixteen TC-rich repeat elements (defense and stress response elements).

As shown in Figure 5, fifty-three of the sixty *PbWRKY* genes possessed different cis-acting elements, while seven other *PbWRKYs* were not detected due to short sequences upstream of the translation initiation site. These five cis-acting elements had a different distribution across the different *PbWRKY* gene families. Most of the *PbWRKYs* contained cis-elements essential for anaerobic induction (83.02%), followed by cis-elements associated with drought stress (49.06%). There were fewer development-related motifs. The CAT-box only had 29 motifs, and the AuxRR-core, circadian, and GON4 motifs had nearly eight motifs, while the others had nearly two or three motifs. From the results, members of the same group or subgroup shared similar potential functions (Figure 5). For example, none of the *PbWRKY* genes in Group IId had cis-elements related to anoxia-specific induction and low-temperature responses, indicating that this subgroup may not function in this area. Similarly, none of the members of Group III contained enhancer-like elements related to anoxia-specific induction, and therefore, members of this group may not respond to anoxic stress (waterlogging stress, etc.). Members of Group IIb lacked the MYB binding site elements involved in drought induction, implying that members of this group may not respond to drought stress. This indicates that *PbWRKYs* are responsive to a variety of abiotic stresses.

### 2.7. Protein Tertiary Structure Prediction

To further validate the results for the different groupings of *PbWRKYs*, the protein tertiary structure of *PbWRKYs* was predicted using the SWISS-MODE online software (https://swissmodel.expasy.org/ (accessed on 25 June 2024)). Each final structure was the most predicted structure in its respective grouping (Figure 6). The results indicated that the tertiary structure of the proteins in each group or subgroup differed significantly. The members of the different subgroups all had similar proportions to those of the corresponding structures, representing a group or subgroup with similar protein structures that may perform similar functions. *PbWRKY36* in Group I had 90% homology (Seq Identity) to *AtWRKY4,* and the value of the qualitative model energy analysis (QMEAN) was 0.59; *PbWRKY46* in Group IIa had 80.30% homology with *AtWRKY18,* and the value of QMEAN was 0.61; *PbWRKY38* in Group IIb had 58.18% homology with *AtWRKY4,* and the value of QMEAN was 0.49; *PbWRKY20* in Group IIc had 66.20% homology with *AtWRKY4,* and the value of QMEAN was 0.58; *PbWRKY17* in Group IId had 55.71% homology with *AtWRKY1,* and the value of QMEAN was 0.67; *PbWRKY45* in Group IIe had 62.34% homology with RRS1, and the value of QMEAN was 0.66; *PbWRKY41* in Group III had 57.81% homology with *OsWRKY45,* and the value of QMEAN was 0.55. *PbWRKY* had the highest homology with *AtWRKY4* and was most likely a homologous protein.

### 2.8. PbWRKY Protein Interaction Network Prediction

To understand the potential regulatory network between *PbWRKY* proteins, a predicted protein interaction network consisting of 26 *PbWRKY* proteins was constructed from 60 *PbWRKYs* using STRING based on homologs of *A. thaliana* (Figure 7). Among the twenty-six *PbWRKY* proteins, three genes pertained to Group I, 19 genes pertained to Group II, and four genes pertained to Group III. The 26 nodes were divided into two categories, one of which had only four WRKY proteins that may be collectively associated with a simple regulatory pathway. *AtWRKY40* (*PbWRKY9*, *PbWRKY46*, and *PbWRKY52*) was located at the center of the regulatory network and had the highest degree of connectivity. It interacted with *AtWRKY18* (*PbWRKY47*) and *AtWRKY60* (*PbWRKY44*), which together regulate ABA signaling. In addition, the interactions of *AtWRKY22* (*PbWRKY16* and *PbWRKY59*) and *AtWRKY33* (*PbWRKY34* and *PbWRKY35*) are associated with hypoxic adaptation to waterlogging stress. *AtWRKY53* (*PbWRKY8* and *PbWRKY23*) plays a significant role in leaf senescence and is also central to stress responses. *AtWRKY70* (*PbWRKY42*) negatively regulates drought resistance by repressing the expression of drought-inducible genes. The homologs of these seven *AtWRKY* counterparts, *PbWRKY*, are also involved in a network of other protein interactions with different *AtWRKY* functions. In summary, these results indicate that there are multiple interactions between *PbWRKY* proteins in response to multiple stresses.

### 2.9. Expression Analysis of the PbWRKYs

Waterlogging and drought stress affect plant growth, and the response to biotic and abiotic stresses occurs mainly at the transcriptional level. To determine whether *PbWRKYs* play a critical role in waterlogging and drought stress responses, we used transcriptome data from two periods (7 and 14 d) under waterlogging stress and three periods (3, 6, and 9 d) under drought stress, as well as rehydration for 3 d after 9 d drought stress, to explore the expression of 60 *PbWRKYs* under waterlogging and drought stress (Figure 8 and Figure 9).

Under drought stress, most *PbWRKY* expression patterns could be classified as either low expression in almost all periods, consistently high expression, or high expression at different periods. Drought treatment for 3, 6, and 9 d, followed by 3 d of rehydration, resulted in 35 (58%), 21 (35%), 24 (40%), and 40 (67%) genes being upregulated compared to the control. *PbWRKY26* and *PbWRKY60* reached the highest expression levels at 3 d with downregulated expression in other periods, and *PbWRKY20* had the highest expression at 9 d. *PbWRKY10*, *PbWRKY22*, *PbWRKY32*, *PbWRKY50*, *PbWRKY51*, and *PbWRKY58* had high expression levels in all periods, while *PbWRKY2*, *PbWRKY4*, *PbWRKY15*, *PbWRKY18*, *PbWRKY21*, *PbWRKY29*, *PbWRKY36*, *PbWRKY38*, and *PbWRKY53* were expressed at low levels in most periods.

The response of *PbWRKYs* under waterlogging stress was not as pronounced as the response to drought stress, with most genes showing a downregulated expression pattern. Their expression profiles under waterlogging stress could be divided into two categories: low expression in most periods and high expression in different periods. *PbWRKY18* and *PbWRKY53* reached their highest expression at 7 d and were subsequently downregulated, while *PbWRKY4*, *PbWRKY49,* and *PbWRKY56* reached their highest expression at 14 d. *PbWRKY2*, *PbWRKY15*, *PbWRKY21*, *PbWRKY29*, *PbWRKY36*, and *PbWRKY38* were consistently highly expressed, which was different from drought stress (Figure 8). This suggests that *PbWRKY* expression patterns differed under waterlogging stress. *PbWRKY10*, *PbWRKY20*, *PbWRKY22*, *PbWRKY26*, *PbWRKY32*, *PbWRKY50*, *PbWRKY51*, *PbWRKY58*, and *PbWRKY60* primarily promoted the *P. bournei* response under drought stress, whereas *PbWRKY2* and *PbWRKY4* mainly promoted the *P. bournei* response under waterlogging stress.

### 2.10. Verification of the Expression of PbWRKYs by RT-qPCR

The expression patterns of 25 selected *PbWRKYs* under drought and waterlogging stresses were examined by RT-qPCR (Figure 10 and Figure 11). Most genes had significant expression after drought and waterlogging stress treatments. Under drought stress, most genes repressed expression in the D2 and D3 periods and upregulated expression in the D4 period (3 d with water after 9 d of drought stress); however, *PbWRKY20*, *PbWRKY50*, *PbWRKY55*, and *PbWRKY57* were upregulated in the D2 and D3 periods; *PbWRKY18*, *PbWRKY36*, and *PbWRKY38* were upregulated in the D2 period and repressed in the D3 period; and *PbWRKY4* repressed expression in the D2 period and regulated expression in the D3 period. The genes with low expression in the D3 period were *PbWRKY23*, *PbWRKY34*, and *PbWRKY43*, and the genes with high expression in the D4 period were *PbWRKY23*, *PbWRKY34*, *PbWRKY35*, *PbWRKY43*, *PbWRKY46*, and *PbWRKY5*.

Some genes were upregulated under waterlogging stress, while others were repressed. *PbWRKY18*, *PbWRKY36*, *PbWRKY38*, *PbWRKY4*, *PbWRKY53*, and *PbWRKY57* were upregulated in the W1 and W2 periods. *PbWRKY11*, *PbWRKY14*, *PbWRKY16*, *PbWRKY22*, *PbWRKY23*, *PbWRKY27*, *PbWRKY31*, *PbWRKY32*, *PbWRKY34*, *PbWRKY35*, *PbWRKY40*, *PbWRKY42*, *PbWRKY43*, *PbWRKY46*, *PbWRKY5*, *PbWRKY50*, *PbWRKY55*, and *PbWRKY*8 were all repressed in the W1 and W2 periods. *PbWRKY20* was almost the same as the CK in the W1 and W2 periods. The gene with the highest expression during the W1 period was *PbWRKY36*; the genes with relatively high expression during the W1 period were *PbWRKY11*, *PbWRKY18*, *PbWRKY34*, *PbWRKY36*, *PbWRKY38*, and *PbWRKY43*. The gene with the highest expression during the W2 period was *PbWRKY*43; the genes with relatively high expression during the W2 period were *PbWRKY4*, *PbWRKY34*, *PbWRKY36*, *PbWRKY38*, and *PbWRKY43*.

In summary, our investigation focused on *PbWRKY23*, *PbWRKY34*, and *PbWRKY43*, specifically observing their responses under drought stress. Similarly, *PbWRKY34*, *PbWRKY36*, *PbWRKY38*, and *PbWRKY43* were the focal points of our examination under waterlogging stress conditions. Notably, *PbWRKY34* and *PbWRKY43* demonstrated distinctive expression patterns, showing specificity under either drought or waterlogging stress, making them particularly intriguing subjects for focused research in these respective environmental conditions.

## 3. Discussion

### 3.1. Different Species and the Number of WRKY Members

The WRKY TF gene family is widespread in woody and herbaceous plants, such as *A. thaliana* [17], *Vitis vinifera* [4], *Triticum aestivum* [10], *Z. mays* [18], and *Nicotiana tabacum* [19] (Table 4). In this study, a search of the WRKY genes in the whole *bournei* genome identified 60 members, which were named *PbWRKY1*-*PbWRKY60* (Table 3). *Phoebe bournei* has fewer WRKY members than *Lycopersicon esculentum* (81) [20], *Osmanthus fragrans* (154) [21], and *Nicotiana tabacum* (164) [19]. However, the number of WRKY genes of *P. bournei* is similar to that of *Santalum album* (64) [22], *V. vinifera* (59) [4], and *Prunus persica* (58) [5], which may be related to the size of the genome. Studies have shown that the quantity of WRKY genes in herbaceous plants is higher than in woody plants, which may be related to the different growth environments of species [10,23,24].

### 3.2. Construction of Phylogenetic Tree and Multiple Sequence Alignment of PbWRKYs

According to the gene structure and conserved motif analysis, *PbWRKYs* could be classified into three groups, namely, Groups I, II, and III, among which Group II could be further divided into five subgroups (IIa–IIe). The numbers of the 60 *PbWRKYs* differed among the different groups. Groups I, II, and III contained thirteen, forty, and seven proteins, respectively. The number of WRKY groups was different in different tree species, such as *A. thaliana* [17] and *Populus tomentosa* [25]. The number of WRKY groups was largest in Group I, while the number of WRKY groups in *O. sativa* was previously reported to be largest in Group III [12]. However, the number of WRKY groups in *P. bournei* has been reported to be largest in Group II, which is the same in *Eucommia ulmoides*, *O. fragrans*, and *Solanum lycopersicum* [21,26,27], indicating that more gene duplications have occurred in Group II during evolution. As shown in Figure 2, the *PbWRKY* proteins of Groups IIa and IIb, as well as IId and IIe, were located close to each other. Therefore, Group II could be divided into five subgroups according to the phylogenetic relationship: IIa, IIb, IIc, IId, and IIe.

The conserved motifs of *PbWRKY* proteins were then evaluated. As shown in Figure 4, the *WRKY*GQK motif was highly conserved in *PbWRKY* proteins, but slight variations were found in four genes (*PbWRKY19*, *PbWRKY29*, *PbWRKY39*, and *PbWRKY44*). The *WRKY* motif of *PbWRKY19* was replaced by WKKY, the same sequence variation occurred in *PbWRKY44*, and the domains of *PbWRKY29* and *PbWRKY39* in Group IIc had deletions. It was previously found that variants of the *WRKY*GQK motif may affect the selection and regulation of the WRKY TF target genes [28,29], and therefore, the function and binding specificity of *PbWRKY19*, *PbWRKY29*, *PbWRKY39*, and *PbWRKY44* are worthy of further research.

The variation in the structural domain is the divergent force of the expansion and evolution of the *PbWRKY* gene family. Variations in the WRKY domain are common in many species, such as *E. ulmoides* and *Cinnamomum camphora* [27,30]. There were large variations in the N-terminal domain of WRKY, which could affect its binding activity to DNA targets. However, the *PbWRKY19* protein in Group I in *P. bournei* displayed a domain loss event, and it was located close to the N-terminus. *PbWRKY29* and *PbWRKY39* in Group IIc and *PbWRKY44* in Group III also had domain loss events. *WRKY*GQK at the conserved site of *PbWRKYs* became WKKYGQK, and it has been speculated that this variation may produce some new biological functions [31,32].

### 3.3. Protein Conserved Motifs and Gene Structure

According to the protein motif composition, *PbWRKYs* were relatively well conserved, with multiple conserved motifs. Motifs 1, 2, 5, 8, 14, and 16 appeared in pairs, indicating that they were functionally related to subgroup classification; motifs 14 and motif 17 were distributed irregularly across the seven subgroups. In general, members of the same subgroup shared common motifs. In terms of gene structure, all members of Group IId contained three exons, and the gene structure was conserved; however, different groups had different numbers of exons. For example, most members of Group I had seven exons, and members of Group III had a maximum of three exons, and therefore, the gene structure was relatively simple. The members of Group II had a maximum of seven to eight exons, and therefore, the gene structure was more complex than that of Groups I and III. This distribution of quantities is similar to that of grape and wheat [10,33].

### 3.4. Ka and Ks Evolutionary Selection

Gene duplication events are one of the reasons why different species have different numbers of WRKY gene families. Gene duplication leads to the evolution of WRKY genes, which may result in new functions, thus improving plant stress resistance [34,35]. According to the duplication sequences of *PbWRKY*, the *K*a/*K*s mutation ratios of 21 pairs of *PbWRKY* duplicated genes were calculated (Table 2). This ratio is an important indicator of study selection limitations and can be used to estimate the approximate timing of duplication events. The *K*a/*K*s ratio of 21 pairs of *PbWRKYs* was <1, which means that the *PbWRKYs* eliminated deleterious mutation sites through purification selection during evolution and that *PbWRKYs* were highly conserved. Additionally, a total of six duplication events were found in 60 *PbWRKYs* and 66 *AtWRKYs* (Table 3), of which five pairs of *K*a/*K*s ratios were <1, and only *PbWRKY37* and *AtWRKY34* had *K*a/*K*s ratios >1, indicating positive selection between these two genes [36,37].

### 3.5. Protein Tertiary Structure Prediction and Protein Regulatory Networks

The protein regulatory network of the *PbWRKY* gene family was investigated. In the network, the colored spheres (protein nodes) were used as optical aids to represent the various input proteins and the forecasted interactions. The protein nodes demonstrated the accuracy of the protein three-dimensional model information. Gray lines connect the proteins, which are related to the circular text evidence. The strength of the data support was indicated by the thickness of the line [38,39]. *AtWRKY40* (*PbWRKY9*, *PbWRKY46*, and *PbWRKY52*) was at the center of the regulatory network and had the most sides, interacting with *AtWRKY18* (*PbWRKY47*) and *AtWRKY60* (*PbWRKY44*), which together regulate ABA signaling [40,41]. Additionally, the interactions of *AtWRKY22* (*PbWRKY16* and *PbWRKY59*) and *AtWRKY33* (*PbWRKY34* and *PbWRKY35*) are associated with hypoxic adaptation to waterlogging stress. *AtWRKY53* (*PbWRKY8* and *PbWRKY23*) regulates leaf senescence and is essential for responding to abiotic stresses such as drought [11]. At the same time, *AtWRKY70* (*PbWRKY42*) negatively regulates drought resistance by repressing the expression of drought-inducible genes [42], and the interactions of *PbWRKY16*, *PbWRKY59*, *PbWRKY34*, *PbWRKY35*, *PbWRKY8*, and *PbWRKY23* may be involved in the mechanism of the waterlogging stress response. There were no obvious interactions between these proteins, and a further correlation analysis may be needed to determine if any exist.

### 3.6. Promoter Cis-Acting Elements

Cis-acting elements in the promoter region affect gene expression by binding to TFs [43]. The analysis results indicated the number of light response elements in the *PbWRKY* promoter, followed by hormone response elements and then stress response elements. A total of 60 *PbWRKY* genes contained 111 G-boxes, which were related to plant hypoxia. They also included hundreds of ABRE response elements and 84 ARE response elements, which were related to anaerobic induction. Finally, they also contained 30 MBS response elements, which are related to the resistance of plants to drought stress. These may respond to drought and waterlogging stress. This discovery suggests that *PbWRKYs* play an important role in the adaptation of *P. bournei* to environmental deterioration, such as waterlogging and drought.

### 3.7. The Expression Pattern of PbWRKYs

The expression pattern of *PbWRKYs* under waterlogging and drought stress was analyzed through the use of heatmaps. An RT-qPCR method was used to verify 25 genes, but only *PbWRKY4*, *PbWRKY18*, *PbWRKY20*, *PbWRKY22*, *PbWRKY23*, *PbWRKY36*, *PbWRKY38*, *PbWRKY40*, *PbWRKY50*, *PbWRKY53*, *PbWRKY55*, and *PbWRKY57* had significant expressions under drought and waterlogging stresses.

Under long-term drought stress, the expression levels of each gene were different on different drought days. The expression of *PbWRKY18*, *PbWRKY22*, *PbWRKY23*, *PbWRKY40*, *PbWRKY50*, and *PbWRKY55* increased markedly under the 3 d drought treatment. After 6 d of drought, *PbWRKY18*, *PbWRKY50*, and *PbWRKY55* remained upregulated; *PbWRKY20*, *PbWRKY38*, and *PbWRKY57* started to be upregulated; and *PbWRKY22*, *PbWRKY23*, and *PbWRKY40*, which were upregulated during D1, started to decrease. At 9 d of drought, *PbWRKY5*, *PbWRKY55*, *PbWRKY5*, *PbWRKY20*, and *PbWRKY38* remained upregulated, while the remaining five genes, *PbWRKY*18, *PbWRKY22*, *PbWRKY23*, *PbWRKY40*, and *PbWRKY50,* started to decrease. *PbWRKY55* remained upregulated during drought and had a significant resistance effect, while all nine genes underwent upregulation under drought treatment. *PbWRKY55*, *PbWRKY18*, *PbWRKY50*, *PbWRKY22*, *PbWRKY23*, and *PbWRKY40* played the most important roles in drought stress, followed by *PbWRKY20*, *PbWRKY38*, and *PbWRKY57*.

The six genes *PbWRKY18*, *PbWRKY36*, *PbWRKY38*, *PbWRKY4*, *PbWRKY53*, and *PbWRKY57* displayed a gradually increasing expression trend under the 7 d waterlogging stress, with the highest expression under the 14 d waterlogging stress. The highest expression was found for *PbWRKY36*. These six genes were the most sensitive to waterlogging treatments. *PbWRKY18*, *PbWRKY20*, *PbWRKY22*, *PbWRKY23*, *PbWRKY38*, *PbWRKY40*, *PbWRKY50*, *PbWRKY55*, and *PbWRKY57* may play a major role in drought stress. *PbWRKY18*, *PbWRKY36*, *PbWRKY38*, *PbWRKY4*, *PbWRKY53*, and *PbWRKY57* played a significant role in the waterlogging treatment, while *PbWRKY18*, *PbWRKY38*, and *PbWRKY57* played positive regulatory roles under both the waterlogging and drought treatments.

The protein interaction network displayed the same trend as in the transcriptome data. *PbWRKY46*, *PbWRKY16*, *PbWRKY34*, *PbWRKY35*, *PbWRKY8*, and *PbWRKY42* decreased significantly under both drought and waterlogging stress. *PbWRKY23* increased first and then decreased under drought stress and decreased significantly under waterlogging stress.

### 3.8. Significantly Expressed PbWRKYs under Drought and Waterlogging Stresses

During drought stress in *P. bournei*, *PbWRKY55* expression was upregulated during D1, D2, and D3, and therefore, it may play a major role in drought stress in *P. bournei*. *AtWRKY61* was located on the same branch as *PbWRKY55*, with both belonging to Group IIb. Jiang showed that *AtWRKY61* expression was significantly increased after drought, cold, and ABSA, which was consistent with the *PbWRKY* transcriptome. *AtWRKY61* may respond to nonbiological stress by interacting with *AtWRKY9* [24,44], and *PbWRKY55* was in the same position as *AtWRKY9* in the protein interaction network. We, therefore, speculated that *PbWRKY55* is an important drought-regulating gene in *P. bournei*.

*PbWRKY36* was consistently upregulated and expressed at high values in W1 and W2 during waterlogging stress in *P. bournei*. We, therefore, speculated that *PbWRKY36* plays a significant role in waterlogging stress in *P. bournei*. *AtWRKY4* is located on the same branch as *PbWRKY36*, and both belong to Group I. Most *PbWRKYs* were similar to *AtWRKY4* in the predicted tertiary structure model of *P. bournei* proteins. Previous research has shown that *AtWRKY4* is a vital salt stress gene; Li and Peng showed that *AtWRKY4* was overexpressed under salt and Me-JA stresses. The role of the *AtWRKY3* and *AtWRKY4* genes in plant defense responses to necrotrophic pathogens has been reported [45]. Roots gradually decay and become necrotic under waterlogged conditions, and we, therefore, speculated that *PbWRKY36* promotes defense responses through the upregulation of root necrosis under waterlogging stress.

*PbWRKY18*, *PbWRKY38*, and *PbWRKY57* had positive regulatory roles under both drought and waterlogging stresses. *AtWRKY45* is homologous to *PbWRKY18*, and previous studies have reported that it displays a strong response to jasmonate treatment [46]. *AtWRKY45* also responds strongly under low-phosphorus stress. *AtWRKY6* is homologous to *PbWRKY38*, and the expression of *AtWRKY6* is affected by the triggering of senescence and plant defense response signals. *AtWRKY6* overexpression has been observed in the presence of low levels of phosphorus, and eight potential target genes have been identified, including 14-3-3 protein, tryptophan synthase, carbonic anhydrases, ATP synthase, and Gln synthetase [47]. *AtWRKY15* is homologous to *PbWRKY57* and regulates plant growth and salt stress responses [48]. *CsWRKY7*, *AtWRKY7*, and *AtWRKY15* are homologous genes. *CsWRKY7* is localized to the nucleus and is involved in the plant response to environmental stress and growth [49]. Therefore, we hypothesized that *PbWRKY18*, *PbWRKY38*, and *PbWRKY57* are all important defense genes in *P. bournei*.

### 3.9. Different Transcription Factors under Drought and Waterlogging Stresses

Transcription factors can regulate the expression of multiple stress-related genes to improve plant stress resistance. Therefore, by overexpressing plant-specific transcription factors in plants, they can simultaneously regulate multiple functional genes and thus improve plant stress resistance, which has become a research focus in genetic improvement and breeding of plant stress resistance. WRKY, MYB, and AP2/ERF are the main transcription factors related to drought and flood resistance in plants.

The WRKY gene family is one of the larger gene families in plants, and the WRKY gene can specifically bind to the downstream gene promoter cis-acting element, which contains a very conserved (C/T) TGAC (T/C) sequence, so it is called W-box. The WRKY gene binds to this element, thereby activating or inhibiting the expression of downstream genes. Drought stress and waterlogging are the most common abiotic stress factors suffered by plants. Plants can regulate their resistance to stress through the WRKY gene regulatory network and hormone signal transduction. The overexpression of the GhWRKY6-like gene in cotton can enhance the tolerance of cotton to drought and osmotic stress, while the MdWRKY30 gene can enhance the tolerance of overexpressed apple callus to salt and osmotic stress by regulating the expression of downstream related genes. It was also found that Arabidopsis thaliana had stronger salt tolerance and osmotic ability than the wild type. In addition, the WRKY gene can also regulate the content of osmotic regulatory substances, such as soluble sugar and proline, and reduce the content of malondialdehyde so as to improve the drought resistance of plants.

MYB is one of the largest transcription factor families in plants. All genes in this family contain a specific conserved domain. The MYB domain consists of 1–4 tandem and non-repetitive R motifs. Numerous studies have shown that MYB genes are involved in abiotic stress pathways and resist stress through their transcriptional regulatory networks. MYB gene is also directly involved in the regulation of external environmental signals. MYB plays an important role in drought stress, and *PtoMYB170* plays a role in the regulation of secondary wall synthesis. Meanwhile, drought experiments in *Arabidopsis Thaliana* showed that the *PtoMYB170* gene plays a regulatory role in drought stress. In addition, it was found that the *MYB156* and *MYB221* genes of poplar, the *AtMYB14* gene of *Arabidopsis Thaliana*, the *MdMYB121* gene of apple, and the *ApMYBs* gene of *Andrographis andrographis* were involved in the regulation of drought stress.

AP2/ERF gene is a transcription factor unique to plants, and its family genes are numerous and versatile. According to the sequence similarity and the number of AP2/ERF functional domains, the AP2/ERF gene family can be divided into AP2, RAV, ERF, and Soloist subfamilies, and the ERF subfamily can be further divided into DREB and ERF subfamilies. DREB gene regulates the expression of related genes by binding to DRE elements in the downstream gene promoter region, the ERF gene activates downstream gene expression by binding to GCC-box elements in the downstream target gene promoter region, and RAV and ERF subfamily genes participate in plant hormone response and abiotic stress response. At present, a large number of AP2/ERF genes have been identified in Arabidopsis Thaliana and rice, and gene function analysis has found that AP2/ERF genes are involved in abiotic stress responses such as waterlogging and drought. *BpERF13* overexpressed transgenic strains of Betulae alba were subjected to cold stress, and it was found that SOD, POD, and CBF genes of transgenic strains were significantly upregulated, indicating that transgenic strains were more cold-resistant. The upregulated expression of *AcERF74* in kiwifruit is involved in the waterlogging response of kiwifruit, and the upregulated expression of *ZmEREB102* in maize is involved in the drought resistance response of maize. In summary, AP2/ERF gene family members are widely involved in abiotic stress responses such as waterlogging, salt, and drought.

## 4. Materials and Methods

### 4.1. WRKY Gene Family Members in P. bournei

The WRKY proteins of *A. thaliana* were downloaded from The Arabidopsis Information Resource (TAIR) website (https://www.arabidopsis.org/ (accessed on 12 April 2023)). Genome Warehouse’s Hidden Markov Model (HMM) (PF03106) was used to identify potential WRKY gene family members (E-value < 10^−5^) [50,51]. Multiple sequence alignment by MEGA was used to exclude redundant fragments [52]. Then, the candidate gene sequences were confirmed in the National Centre for Biotechnology Conserved Domains Database (NCBI-CDD) (https://www.ncbi.nlm.nih.gov/cdd/ (accessed on 20 April 2023)) and the Pfam database (http://pfam-legacy.xfam.org/ (accessed on 9 May 2023)) [51]. ExPASy prediction (https://web.ExPASy.org/protparam/ (accessed on 24 May 2023)) was used to obtain the physicochemical properties of the completed *PbWRKY* gene family members, which included molecular weight, isoelectric point, number of amino acids, instability coefficient, aliphatic index, hydropathicity, and predicted location.

### 4.2. Multiple Sequence Alignment and Phylogenetic Analysis

The phylogenetic relationship among the aligned sequences obtained was estimated by the neighbor-joining (NJ) method using the MEGA 7.0 software [52]. Further, phylogenetic tree beautification was performed by Evolview 8 (https://www.evolgenius.info/evolview (accessed on 7 June 2023)). Ultimately, the protein evolutionary tree consisted of the WRKY family members of *A. thaliana* and *P. bournei*.

### 4.3. Gene Structures and Conserved Motif Analysis

The PlantCARE database (https://bioinformatics.psb.ugent.be/webtools/plantcare/html/ (accessed on 12 June 2023)) and TBtools [53] were used to predict the structure of *PbWRKY* introns and exons. To identify the motif structure of the conserved domain of *PbWRKYs*, an online MEME was used (https://meme-suite.org/meme/doc/meme.html (accessed on 12 June 2023)) [54].

### 4.4. Calculation of Non-Synonymous (Ka) and Synonymous (Ks) Values

Non-synonymous and synonymous substitution events were estimated using *K*a/*K*s values, and *K*s and *K*a replacement rates were calculated using the Nei and Gojobori methods implemented in the *K*a*K*s calculator (http://code.google.com/p/kaks-calculator/wiki/KaKs_Calculator (accessed on 15 June 2023)) [55].

### 4.5. Promoter Cis-Regulatory Element Analysis

The start codon of the *P. bournei* genome was located by PlantCARE (https://bioinformatics.psb.ugent.be/webtools/plantcare/html/ (accessed on 24 June 2023)), and the 1500-bp sequence of its upstream region was intercepted as the gene promoter sequence. The results were processed using Excel software (2016), followed by TBtools software (v2. 056) for visualization [53].

### 4.6. Protein Tertiary Structure Prediction and Interaction Network Analysis

The protein tertiary structure was predicted using SWISS-MDEL (https://swissmodel.expasy.org/interactive (accessed on 27 June 2023)), and the highest percentage of protein tertiary structure models was found for each group. Using Ortho Venn 2 (https://orthovenn2.bioinfotoolkits.net/home (accessed on 28 June 2023)) and String (https://string-db.org/ (accessed on 29 June 2023)), the WRKY protein sequences of the two species were compared, and protein interaction networks were mapped.

### 4.7. Plant Materials and Abiotic Stress Treatments

Three hundred healthy *P. bournei* seedlings (three years old) were obtained from the Fujian Academy of Forestry Sciences and then transplanted to the Fujian Agriculture and Forestry University nursery for drought and waterlogging stress experiments. Place 300 young seedlings in a greenhouse, and for the seedlings under drought stress, no water is provided directly. For the seedlings under waterlogging stress, a large pot is placed inside a smaller pot so that the roots of the seedlings are fully submerged in water. Drought and waterlogging stress occurred between 29 July and 25 August 2021. Prior to 29 July, we initiated several drought stress experiments. However, due to weather conditions, i.e., the start of a rain event after 3 d of drought, we had to terminate the experiment. There was no rain from 29 July to 10 August. According to previous experiments, the seedlings were inactivated for 12 d under drought and 35 d under waterlogging stress. We took photographs and root samples after treatments of 0, 3, 6, and 9 d under drought stress and 3 d after rewatering. The post-stress recovery was suitable for drought-affected plants. During the pre-experiment period, we conducted recovery experiments on plants under waterlogging stress for 28 d, but at this time, the seedlings had already become inactive and could not be measured for transcriptome and physiological indicators. We took photographs and root samples after waterlogging treatments of 0, 7, 14, 21, and 28 d. Each sampling used the same procedure and was repeated five times, and the whole plant, as well as its roots and leaves, were photographed with a Leica M Type 240 camera (Appendix A). Generally, mature trees have better tolerance to environmental stress. Field observations have shown that seedlings are more likely to die when drought or waterlogging occurs. Because this was a controllable stress test, small seedlings were easy to work with. Currently, only the response characteristics of the seedling stage to stress are understood, and there are no clear results for the study of older trees.

### 4.8. RT-qPCR Analysis

Each sample was weighed to 0.1−0.2 g and ground rapidly in liquid nitrogen. The total RNA of *P. bournei* was extracted from different periods using a TIANGEN RNA prep Pure (DP441) Polysaccharide and Polyphenol Plant Total RNA Extraction Kit (centrifuge column), which was designed to prevent RNase contamination. The integrity of the RNA was checked using 1% agarose gel electrophoresis, and the concentration and purity were measured by ultraviolet spectroscopy. RNA was used for cDNA synthesis.

cDNA was synthesized using the NovaBio HyperScript^TM^ III RT Super Mix for RT-qPCR with the gDNA Remover Kit. Genomic DNA was removed to constitute a 20 μL reaction system (Appendix A). RG6, RG7, and RG8 were used as internal reference genes [56]. The RT-qPCR technique was used to examine the expression of 11 internal reference genes at seven different times in the roots of *P. bournei*. The NovaBio 2*S6 Universal SYBR RT-qPCR Mix kit (operated on an Icycler iQ5 (BioRad, Hercules, CA, USA)) was used for this analysis. The total reaction system was 20 μL in a 96-well plate (Appendix A), and three repeated operations were performed for each sample.

## 5. Conclusions

We studied the evolution of the WRKY gene family in *P. bournei* and other species and systematically identified the *PbWRKY* gene family in *P. bournei* at the genome-wide level. A total of 60 *PbWRKY* genes were identified and classified into three groups. These genes had similarities and differences in physicochemical properties, gene structure, conserved motifs, cis-regulatory elements, gene duplication, and the protein interaction network. Transcriptome data indicated that *PbWRKY4*, *PbWRKY18*, *PbWRKY20*, *PbWRKY22*, *PbWRKY23*, *PbWRKY36*, *PbWRKY38*, *PbWRKY40*, *PbWRKY50*, *PbWRKY53*, *PbWRKY55*, and *PbWRKY57* had significant expression under drought and waterlogging stresses. The results serve not only as a foundation for the selection of alternative genes for abiotic stress responses but will also inform future studies of the effects of WRKY in woody plants.

## Figures and Tables

**Figure 1 ijms-25-07280-f001:**
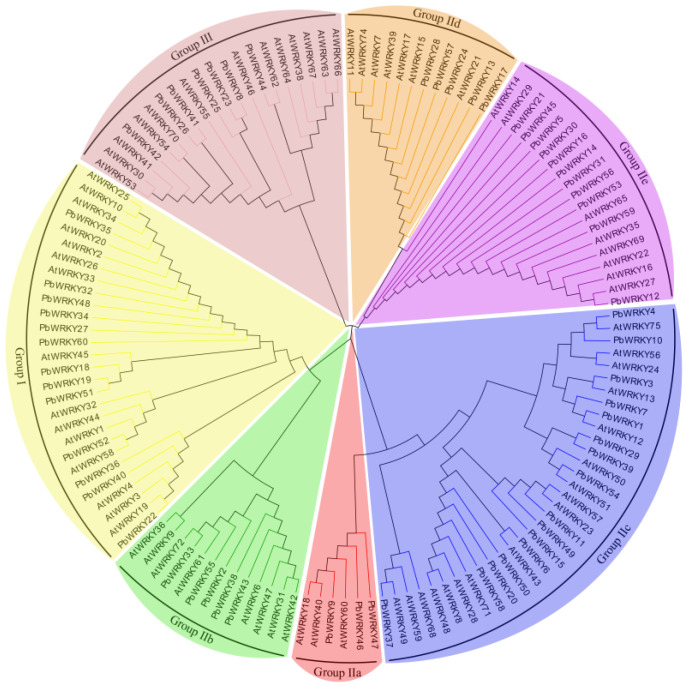
Phylogenetic tree of WRKY proteins from *P. bournei* and *Arabidopsis thaliana*. The tree was divided into seven phylogenetic groups distinguished by color. The colors yellow, red, green, blue, orange, purple, and pink represent Groups I, IIa, IIb, IIc, IId, IIe, and III, respectively. The names of the subfamilies are shown outside of the circle.

**Figure 2 ijms-25-07280-f002:**
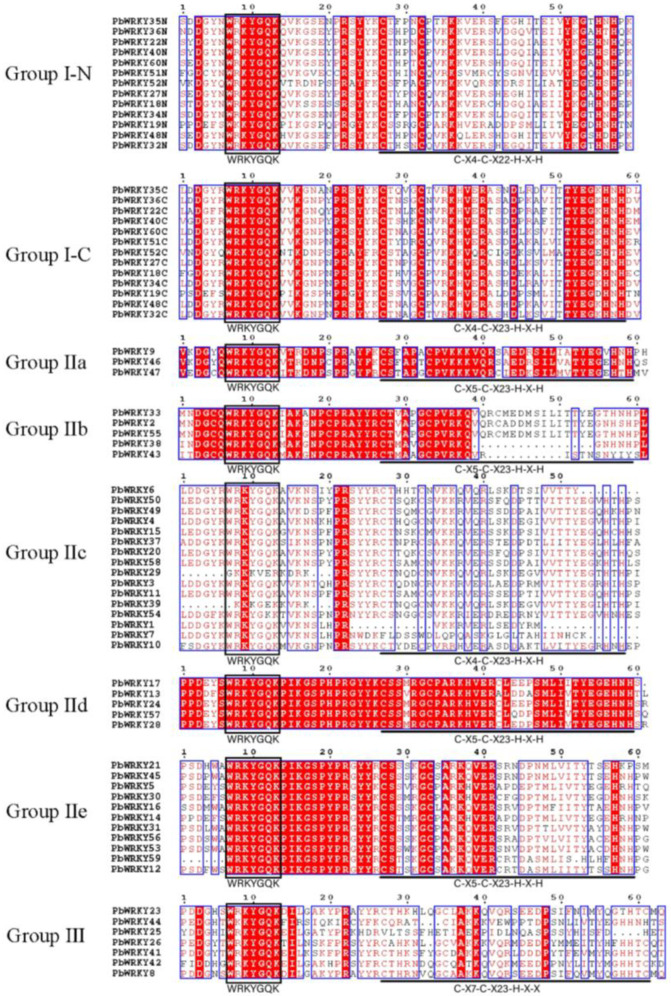
Multiple sequence alignment of the WRKY domains of 60 *PbWRKY* proteins. I-N, I-C, II-a, II-b, II-c, II-d, II-e, and III represent different subgroups in three subfamilies; colors indicate sequence similarity; and a darker color indicates higher conservatism.

**Figure 3 ijms-25-07280-f003:**
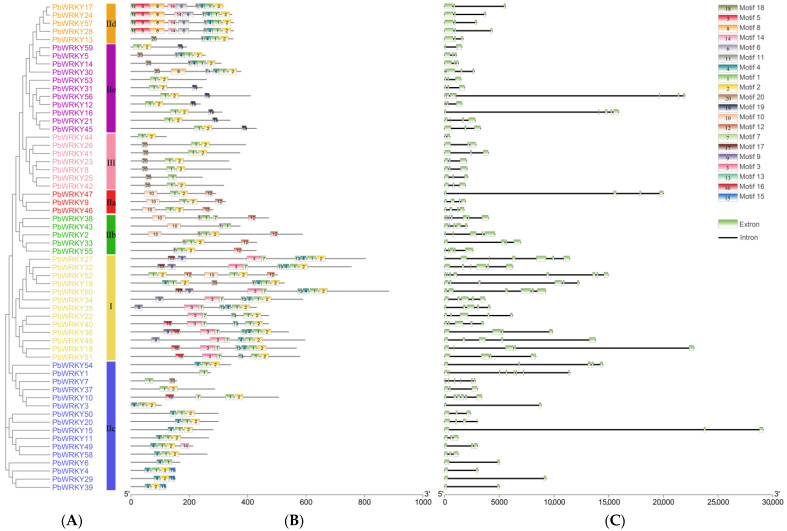
Phylogenetic relationship, gene structure, and motif compositions of *PbWRKYs*. (**A**) An unrooted phylogenetic tree constructed using MEGA7.0. (**B**) Conserved motifs of 60 *PbWRKY* proteins. MEME was used to predict motifs, which are represented by different colored boxes numbered 1–20. (**C**) Exon–intron structure of *PbWRKYs*.

**Figure 4 ijms-25-07280-f004:**
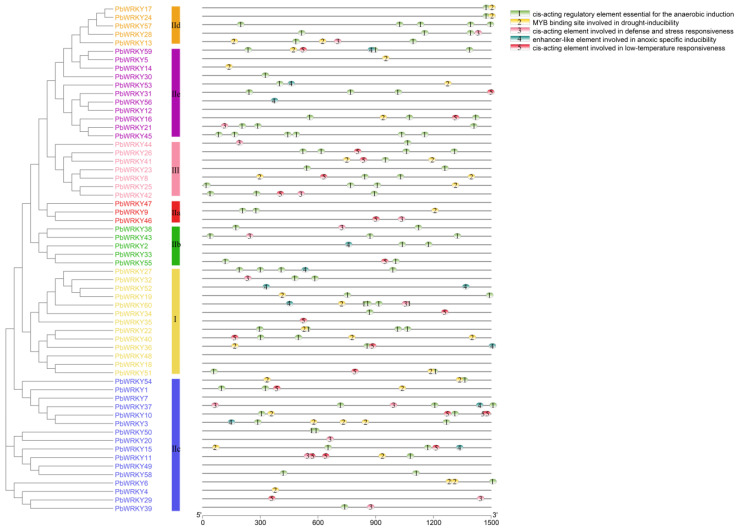
Prediction of cis-regulatory elements in the promoter regions of the *PbWRKY* gene family.

**Figure 5 ijms-25-07280-f005:**
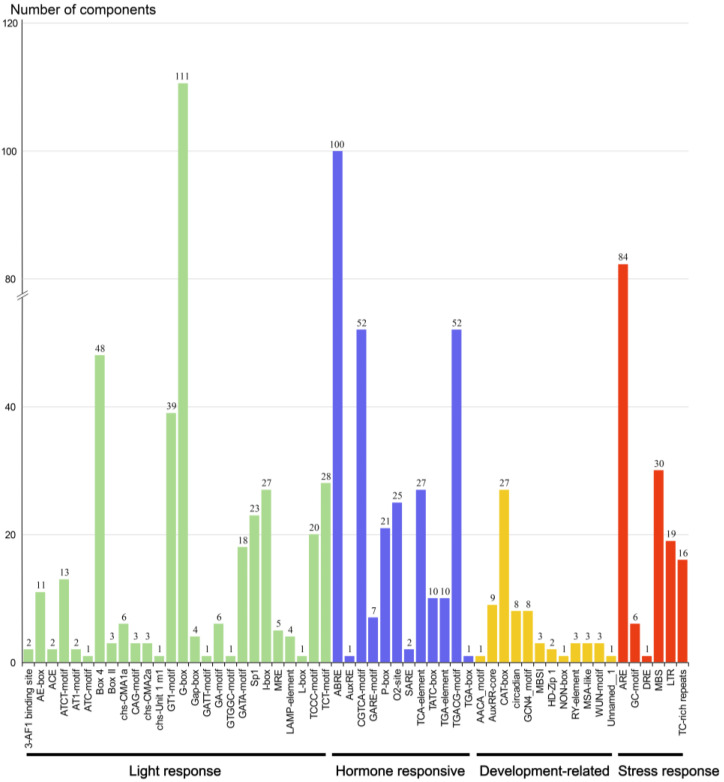
Cis-element analysis in the promoter of *PbWRKY* genes. The cis-element distributed in the promoters of *PbWRKY*. The number of cis-acting elements in the promoter region of each *PbWRKY* gene.

**Figure 6 ijms-25-07280-f006:**
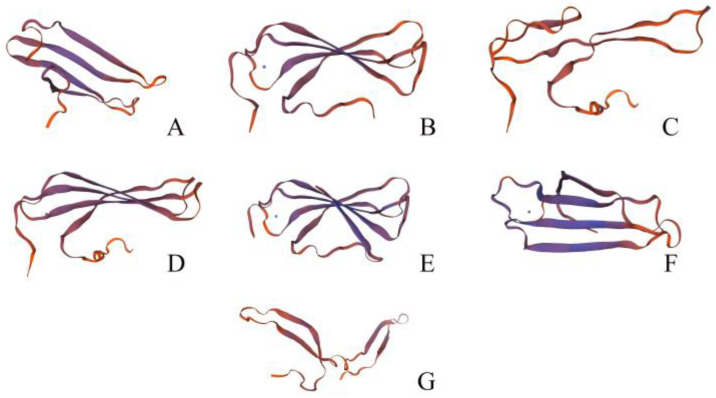
Prediction of protein tertiary structure. (**A**–**G**) represent Groups I, IIa, IIb, IIc, IId, IIe, and III, respectively, and represent the most important protein structure predictions.

**Figure 7 ijms-25-07280-f007:**
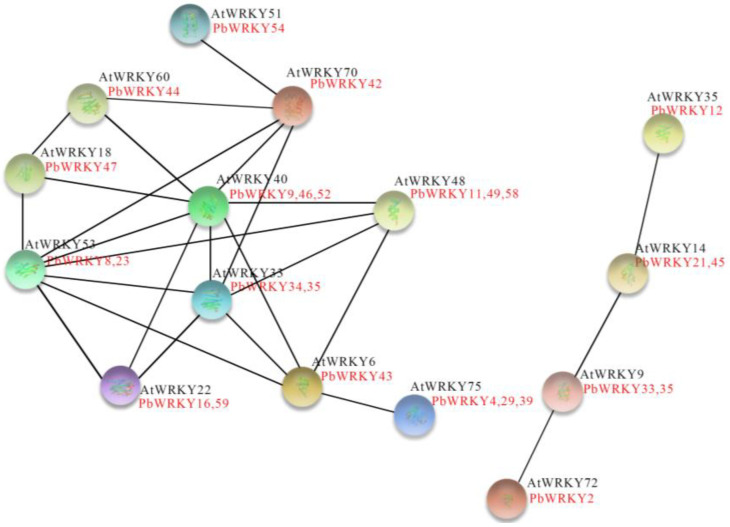
Regulatory networks of the *PbWRKY* gene family. Colored balls (protein nodes) in the network were used as a visual aid to indicate different input proteins and predicted interactions. Protein nodes that are enlarged indicate the availability of three-dimensional protein structure information. Gray lines connect proteins that are associated with recurring text-mining evidence. Line thickness indicates the strength of data support.

**Figure 8 ijms-25-07280-f008:**
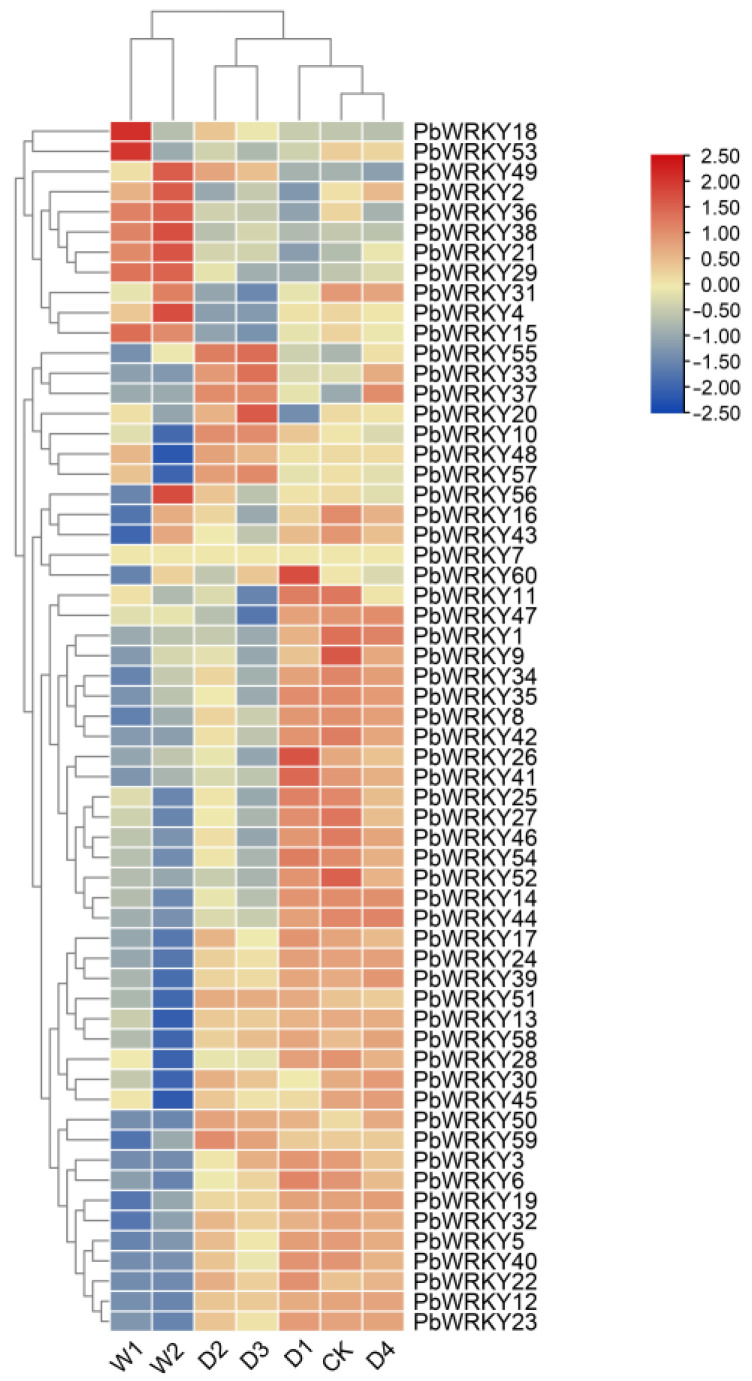
Heatmaps of expression profiles of PbWRKY genes in response to drought and waterlogging. CK represents normal watering; W1 represents 7 d under waterlogging stress; W2 represents 14 d under waterlogging stress; D1 represents 3 d under drought stress; D2 represents 6 d under drought stress; D3 represents 9 d under drought stress; D4 represents 3 d with water after 9 d of drought stress.

**Figure 9 ijms-25-07280-f009:**
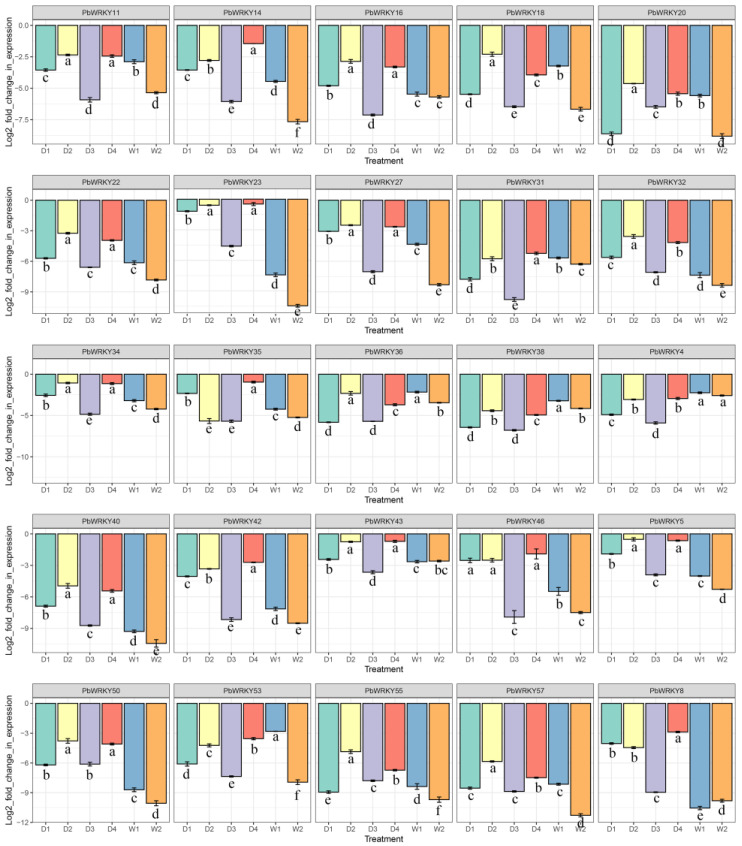
Real-time quantitative PCR (RT-qPCR) expression levels of selected *PbWRKY* genes following drought stress and waterlogging stress. *PbWRKY* gene expression levels were analyzed by RT-qPCR, and RG6, RG7, and RG8 were used as internal reference genes. The relative expression levels were calculated using the 2^−ΔΔCT^ method. Data represent the mean ± standard deviation (*p* < 0.05). To calculate the letters, we use least significant difference (LSD). The error bars a–f represent standard error in figures.

**Figure 10 ijms-25-07280-f010:**
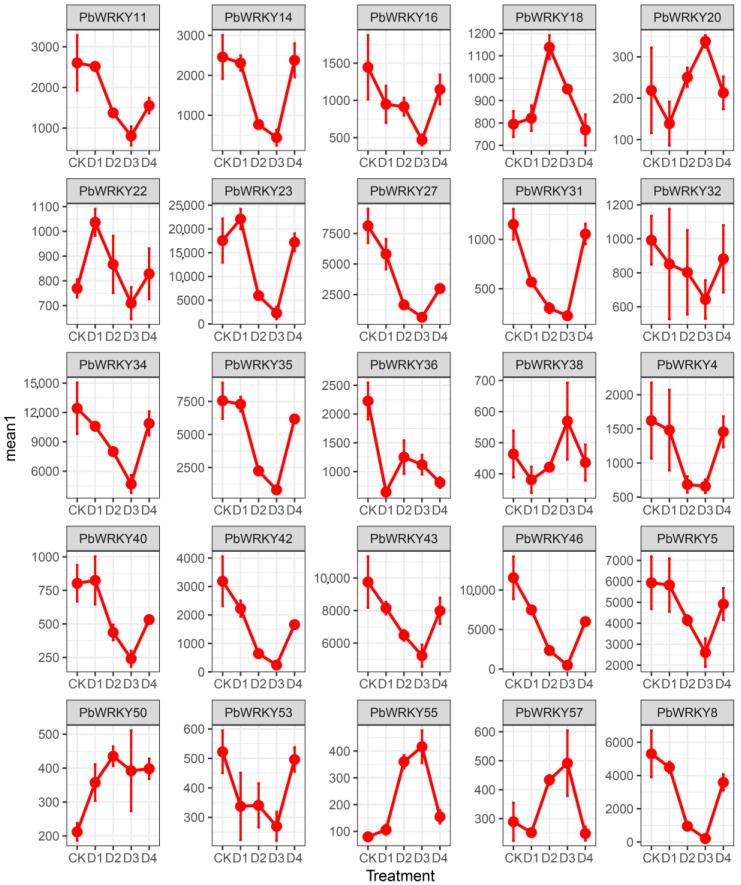
Expression patterns of 25 *PbWRKY* genes under drought stress. Gene expression levels at 0, 3, 6, and 9 d under drought stress and at 3 d with water after 9 d of drought stress.

**Figure 11 ijms-25-07280-f011:**
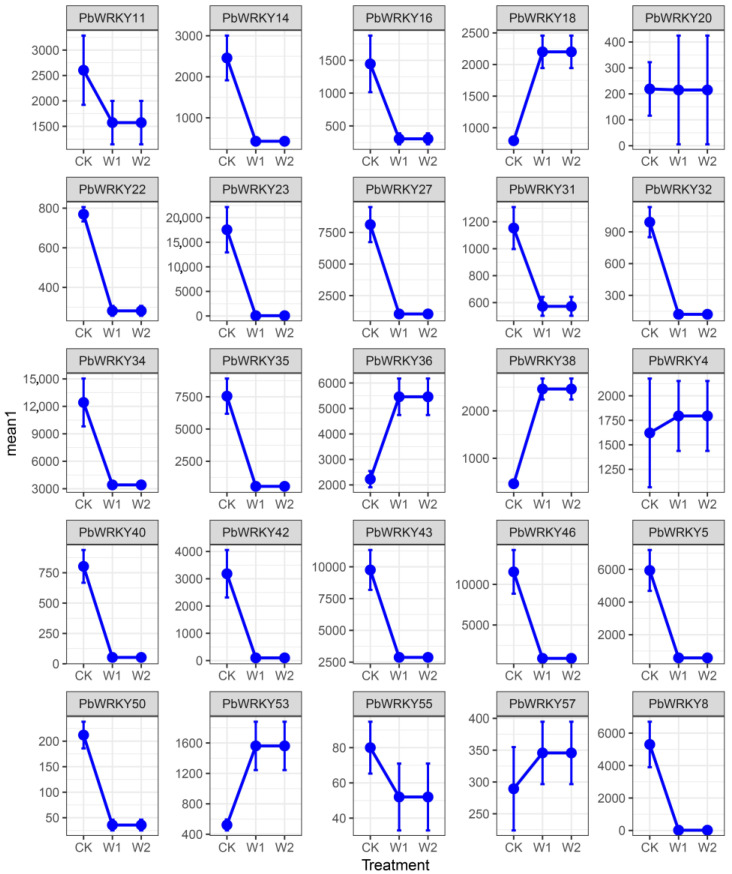
Expression patterns of 25 *PbWRKY* genes under waterlogging stress. Gene expression levels at 0, 7, and 14 d under waterlogging stress.

**Table 1 ijms-25-07280-t001:** Details of information on the WRKY transcription factor family in *P. bournei*.

Gene Name	Gene ID	Group	Number of Amino Acids (aa)	Isoelectric Point	Molecular Weight (kDa)	Instability Coefficient	Aliphatic Index	Hydropathicity	Predicted Location(s)
*PbWRKY1*	Maker00000780	Group IIc	273	9.89	31,244.61	56.27	74.21	−0.728	Nucleus
*PbWRKY2*	Maker00001301	Group IIb	588	5.67	63,553.76	56.5	55.6	−0.775	Nucleus
*PbWRKY3*	Maker00001465	Group IIc	104	9.74	12,638.47	42.46	53.27	−1.159	Nucleus
*PbWRKY4*	Maker00002106	Group IIc	153	9.49	17,700.71	25.57	52.09	−1.174	Nucleus
*PbWRKY5*	Maker00047617	Group IIe	255	9.87	27,413.13	47.73	62.43	−0.567	Nucleus
*PbWRKY6*	Maker00003851	Group IIc	168	8.9	18,891.24	43.26	62.14	−0.776	Nucleus
*PbWRKY7*	Maker00049281	Group IIc	157	4.83	18,160.4	36.46	75.73	−0.533	Nucleus
*PbWRKY8*	Maker00006973	Group III	343	6.08	37,173.18	61.97	55.45	−0.647	Nucleus
*PbWRKY9*	Maker00007836	Group IIa	324	8.65	35,884.63	55.42	78.21	−0.572	Nucleus
*PbWRKY10*	Maker00007878	Group IIc	506	6.99	54,869.19	57.13	65.61	−0.679	Nucleus
*PbWRKY11*	Maker00009559	Group IIc	266	8.67	29,502	63.23	47.63	−0.73	Nucleus
*PbWRKY12*	Maker00050225	Group IIe	238	5.7	27,228.01	55.15	48.66	−1.092	Nucleus
*PbWRKY13*	Maker00010486	Group IId	350	9.68	38,517.92	56.57	62.4	−0.694	Nucleus
*PbWRKY14*	Maker00013669	Group IIe	309	9.53	33,398.89	47.2	64.17	−0.541	Nucleus
*PbWRKY15*	Maker00014626	Group IIc	281	9.11	31,562.24	75.91	50.75	−0.99	Nucleus
*PbWRKY16*	Maker00014633	Group IIe	312	4.8	34,429.01	53.42	58.49	−0.708	Nucleus
*PbWRKY17*	Maker00014832	Group IId	316	9.73	35,494.34	58.22	62.94	−0.769	Nucleus
*PbWRKY18*	Maker00052053	Group I	567	6.17	61,295.1	55.5	62.33	−0.778	Nucleus
*PbWRKY19*	Maker00014965	Group I	526	6.86	58,937.58	56.7	64.89	−0.734	Nucleus
*PbWRKY20*	Maker00015072	Group IIc	299	6.67	32,771.19	53.89	45.69	−0.852	Nucleus
*PbWRKY21*	Maker00015451	Group IIe	340	8.7	37,842.51	51.07	52.79	−0.851	Nucleus
*PbWRKY22*	Maker00015513	Group I	472	9.03	52,455.14	49.45	62.14	−0.906	Nucleus
*PbWRKY23*	Maker00016108	Group III	335	5.61	37,228.27	60.53	55.34	−0.696	Nucleus
*PbWRKY24*	Maker00017029	Group IId	346	9.66	38,811.26	55.42	67.08	−0.67	Nucleus.
*PbWRKY25*	Maker00019100	Group III	245	7.21	27,159.44	55.2	66.98	−0.657	Nucleus
*PbWRKY26*	Maker00019162	Group III	393	5.67	43,235.5	60.32	63.46	−0.649	Nucleus
*PbWRKY27*	Maker00019539	Group I	804	6.27	87,881.38	52.47	65.12	−0.686	Nucleus
*PbWRKY28*	Maker00019925	Group IId	352	9.67	39,482.8	59.07	62.59	−0.718	Nucleus
*PbWRKY29*	Maker00022460	Group IIc	152	9.17	17,118.96	56.09	48.03	−0.932	Nucleus
*PbWRKY30*	Maker00023278	Group IIe	377	9.39	42,276.26	48.94	67.51	−0.551	Nucleus
*PbWRKY31*	Maker00054439	Group IIe	244	4.8	26,674.46	55.26	51.19	−0.71	Nucleus
*PbWRKY32*	Maker00024933	Group I	755	5.86	81,485.91	51.08	57.58	−0.726	Nucleus
*PbWRKY33*	Maker00025694	Group IIb	431	7.58	47,405.23	40.86	69.74	−0.448	Nucleus
*PbWRKY34*	Maker00027250	Group I	589	7.27	64,199.34	59.51	48.05	−0.837	Nucleus
*PbWRKY35*	Maker00027367	Group I	430	8.43	47,970.63	56.4	50.58	−0.925	Nucleus
*PbWRKY36*	Maker00027433	Group I	540	7.68	58,680.95	52.6	58.52	−0.788	Nucleus.
*PbWRKY37*	Maker00028249	Group IIc	287	6.19	31,607.24	61.16	66.55	−0.689	Nucleus
*PbWRKY38*	Maker00028411	Group IIb	472	8.11	51,029.39	52.6	63.35	−0.64	Nucleus
*PbWRKY39*	Maker00030322	Group IIc	121	9.42	13,742.49	18.55	54.63	−1.029	Nucleus
*PbWRKY40*	Maker00031212	Group I	471	8.86	52,043.28	47.34	67.22	−0.77	Nucleus
*PbWRKY41*	Maker00034790	Group III	373	6.12	41,383.36	54.28	64.08	−0.61	Nucleus
*PbWRKY42*	Maker00034795	Group III	318	5.66	35,773.75	62.63	57.99	−0.801	Nucleus
*PbWRKY43*	Maker00035060	Group IIb	374	8.14	41,736.37	47.39	58.37	−0.923	Nucleus
*PbWRKY44*	Maker00035165	Group III	121	8.37	13,706.29	32.85	60.41	−0.855	Nucleus
*PbWRKY45*	Maker00035245	Group IIe	430	5.28	47,093.09	43.73	54.05	−0.794	Nucleus
*PbWRKY46*	Maker00038149	Group IIa	281	8.43	31,372.54	55.78	65.23	−0.775	Nucleus.
*PbWRKY47*	Maker00038153	Group IIa	291	9.55	32,057.29	43.07	89.07	−0.179	Nucleus
*PbWRKY48*	Maker00039321	Group I	596	6.56	64,711.83	52.34	64.41	−0.689	Nucleus
*PbWRKY49*	Maker00039574	Group IIc	213	6.87	24,133.99	58.53	55.31	−1.006	Nucleus
*PbWRKY50*	Maker00039996	Group IIc	299	8.64	32,793.45	59.94	50.9	−0.787	Nucleus
*PbWRKY51*	Maker00040415	Group I	579	6.46	63,327.63	52.31	68.22	−0.646	Nucleus
*PbWRKY52*	Maker00040441	Group I	502	9.16	55,453.72	45.18	70.14	−0.608	Chloroplast Nucleus
*PbWRKY53*	Maker00040627	Group IIe	259	10.8	29,016.71	81.53	50.85	−1	Nucleus
*PbWRKY54*	Maker00042530	Group IIc	343	8.26	38,648.52	52.06	69.88	−0.579	Nucleus
*PbWRKY55*	Maker00044088	Group IIb	429	6.18	47,339.78	42.13	63.73	−0.708	Nucleus
*PbWRKY56*	Maker00045608	Group IIe	410	6.89	45,361.96	60.23	61.61	−0.622	Nucleus.
*PbWRKY57*	Maker00046708	Group IId	353	9.65	39,400.88	62.99	72.41	−0.656	Nucleus
*PbWRKY58*	Maker00047219	Group IIc	261	7.67	29,187.93	64.6	67.62	−0.737	Nucleus
*PbWRKY59*	Maker00047258	Group IIe	191	6.3	21,820.45	52.81	60.21	−0.913	Nucleus
*PbWRKY60*	Maker00047509	Group I	883	6.16	96,252.78	53.78	71.68	−0.523	Nucleus

**Table 2 ijms-25-07280-t002:** *K*a/*K*s analysis and divergence time estimated for *P. bournei* duplicated WRKY paralogs.

Gene Pairs		Ka	Ks	Ka/Ks	Duplication Date (MYA)
*PbWRKY17*	*PbWRKY24*	0.01	0.07	0.15	5.45
*PbWRKY12*	*PbWRKY59*	0.18	0.41	0.45	31.35
*PbWRKY57*	*PbWRKY28*	0.12	0.44	0.27	34.16
*PbWRKY29*	*PbWRKY39*	0.15	0.53	0.29	41.02
*PbWRKY33*	*PbWRKY55*	0.19	0.63	0.31	48.57
*PbWRKY13*	*PbWRKY19*	0.22	0.69	0.32	53.02
*PbWRKY22*	*PbWRKY40*	0.23	0.69	0.34	53.38
*PbWRKY23*	*PbWRKY8*	0.21	0.77	0.27	59.17
*PbWRKY35*	*PbWRKY34*	0.24	0.77	0.31	59.39
*PbWRKY5*	*PbWRKY14*	0.17	0.95	0.18	72.95
*PbWRKY51*	*PbWRKY18*	0.36	0.99	0.36	76.33
*PbWRKY21*	*PbWRKY45*	0.31	1.00	0.32	76.66
*PbWRKY58*	*PbWRKY11*	0.13	1.02	0.13	78.56
*PbWRKY31*	*PbWRKY53*	0.36	1.26	0.29	96.61
*PbWRKY41*	*PbWRKY26*	0.46	1.30	0.35	99.74
*PbWRKY56*	*PbWRKY53*	0.58	1.38	0.42	105.87
*PbWRKY52*	*PbWRKY47*	0.42	1.44	0.29	110.93
*PbWRKY50*	*PbWRKY20*	0.26	1.58	0.17	121.47
*PbWRKY9*	*PbWRKY46*	0.38	1.86	0.21	143.44
*PbWRKY60*	*PbWRKY32*	0.43	2.24	0.19	172.04
*PbWRKY38*	*PbWRKY2*	0.54	3.15	0.17	241.94

**Table 3 ijms-25-07280-t003:** *K*a/*K*s analysis and estimated divergence time of *WRKY* genes from *P. bournei* and *A. thaliana*.

Gene Pairs		Ka	Ks	Ka/Ks	Duplication Date (MYA)
*PbWRKY37*	*AtWRKY34*	3.11	1.82	1.71	140.37
*PbWRKY5*	*AtWRKY29*	0.64	2.23	0.29	171.35
*PbWRKY11*	*AtWRKY45*	0.40	2.44	0.16	187.73
*PbWRKY25*	*AtWRKY63*	0.85	2.65	0.32	203.89
*PbWRKY26*	*AtWRKY54*	0.84	2.70	0.31	207.34
*PbWRKY41*	*AtWRKY30*	0.66	4.03	0.16	309.77

**Table 4 ijms-25-07280-t004:** The number and type of WRKY proteins in 28 plants.

Speices	Group I	Group IIa	Group IIb	Group IIc	Group IId	Group IIe	Group III	Total
*Camellia oleifera*	20	6	10	20	9	11	13	89
*Poplar 84K*	21	5	9	24	13	16	10	98
*Camphora officinarum*	11	4	8	13	11	6	7	60
*Pinus massoniana*	13	10	4	6	6	3	1	43
*Eucalyptus grandis*	17	11	9	15	8	7	12	79
*Casuarina equisetifolia*	10	4	8	16	7	7	12	64
*Amborella trichopoda*	7	2	5	6	3	3	8	34
*Nymphaea colorata*	18	6	9	10	7	10	5	65
*Cinnamomum micranthum*	15	6	8	12	9	12	11	73
*Liriodendron chinense*	7	3	6	8	4	7	9	44
*Liriodendron tulipifera*	9	6	7	13	6	7	10	58
*Brachypodium arbuscula*	13	4	8	23	7	10	26	91
*Brachypodium distachyon*	17	5	5	16	10	10	26	89
*Oryza sativa*	7	4	8	19	6	11	51	106
*Sorghum bicolor*	9	5	8	20	7	11	36	96
*Zea mays*	15	7	12	30	12	17	44	137
*Zostera marina*	7	2	8	10	6	6	7	46
*Arabidopsis thaliana*	12	3	8	18	7	7	28	83
*Carya illinoinensis*	17	6	13	24	9	10	12	91
*Citrus trifoliata*	10	3	8	13	5	7	9	55
*Coffea arabica*	23	5	15	28	13	11	32	127
*Corymbia citriodora*	16	6	11	18	5	8	15	79
*Gossypium hirsutum*	36	16	31	69	28	26	32	238
*Malus pumila*	23	6	15	25	14	13	28	124
*Populus trichocarpa*	23	5	9	24	13	12	14	100
*Passiflora edulis*	14	5	9	10	8	5	4	55
*Sinapis alba*	41	8	26	64	24	19	51	233

## Data Availability

Genome sequences have been submitted to the National Genomics Data Center (NGDC). The raw whole-genome data of *P. bournei* have been deposited in BioProject/GSA (https://bigd.big.ac.cn/gsa (accessed on 25 March 2023)) under the accession codes PRJCA002001/CRA002192, and the assembly and annotation of the whole-genome data have been deposited at BioProject/GWH (https://bigd.big.ac.cn/gwh (accessed on 27 March 2023)) under the accession codes PRJCA002001/GWHACDM00000000.

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
