# Peer review of "Comprehensive Expression Analysis of the WRKY Gene Family in Phoebe bournei under Drought and Waterlogging Stresses"

_ijms, 2024, doi:10.3390/ijms25137280_

Round 1

Reviewer 1 Report (Previous Reviewer 2)

Comments and Suggestions for Authors

Paper is now ready for acceptance after including the requested data. Please mind the typo in "species" in table 4.

Author Response

Comments 1: Paper is now ready for acceptance after including the requested data. Please mind the typo in "species" in table 4.

Response 1: In table 4, we changed " Cinnamomum camphora" into " Camphora officinarum"; changed " Cinnamomum kanehirae" into " Cinnamomum micranthum".

Reviewer 2 Report (New Reviewer)

Comments and Suggestions for Authors

This study focused on Phoebe bournei, and involved a genome-wide identification of WRKY gene family members, clarification of their molecular evolutionary characteristics, and comprehensive mapping of their expression profiles under diverse abiotic stress conditions for the first time.

Some figures and tables are not properly arranged and need to be modified;

1.         Table 1 is too large and it is suggested arranged as a supplementary Table;

2.         Fig.2 There are too many blank spaces around the figure, it is recommended to cut the edge blanks, and strive to compress the plate space;

3.         Fig.3, the figure is not clear, either enlarged or separated into two figs;

4.         Fig. 4 characters are too small, please enlarge the figure. Transplant the fig on the right to the bottom to zoom in on the entire figure so that it can be seen clearly;

5.         Fig. 8. Shrink the figure, the same size as the other figs;

Other problem:

6.         Line 105-106, do not use “may” in the text, you should have a test to show their subcellular localization;

7.         Whether PbWRKY gene was transferred into Arabidopsis and expression analysis.

Comments on the Quality of English Language

no

Author Response

Comments 1: Table 1 is too large and it is suggested arranged as a supplementary Table;

Response 1: Table 1 contains important data from PBWRKY and has been used and discussed many times, so I think it is more intuitive and convenient to put it in the article.

Comments 2: Fig.2 There are too many blank spaces around the figure, it is recommended to cut the edge blanks, and strive to compress the plate space;

Response 2: In Figure 2, the marginal blanks have been cut out.

Comments 3: Fig.3, the figure is not clear, either enlarged or separated into two figs;

Response 3: Figure 3 has been enlarged.

Comments 4: Fig. 4 characters are too small, please enlarge the figure.   Transplant the fig on the right to the bottom to zoom in on the entire figure so that it can be seen clearly;

Response 4: Figure 4 has been enlarged;

Comments 5: Fig. 8.   Shrink the figure, the same size as the other figs;

Response 5: Figure 8 has been scaled down.

Comments 6:  Line 105-106, do not use “may” in the text, you should have a test to show their subcellular localization;

Response 6: In line 105-106, we have removed "may".

Comments 7: Whether PbWRKY gene was transferred into Arabidopsis and expression analysis.

Response 7: I'm sorry that the project has ended, and there will be no more experiments.

This manuscript is a resubmission of an earlier submission. The following is a list of the peer review reports and author responses from that submission.

Round 1

Reviewer 1 Report

Comments and Suggestions for Authors

 The Ms Comprehensive Expression Analysis of the WRKY Gene Family in Phoebe bournei Under Drought and Waterlogging Stresses, needs major revision.

The introduction section should include a para for drought and waterlogging stress, mechanism of sensing and responses in plants etc. These are all well known. Authors may see PMID: 36611920.  Then the authors can move to the studied plants and WRKY Tfs. Moreover, various other TFs has also been reported to be associated with these stresses, that may also be included as background. Numerous TFs role have been recently compiled in a book by Elsevier i.e. plant-transcription-factors.

The method section needs to be further elaborated with experimental and statistical details.

Discussion again should be comparative with the earlier reports. It is seems to be the repetition of the result section in most of the para, and majority of things have not been discussed. What are the other genes involved in these stress, how they functions, how WRKY is different of related to those genes, is their any cross talk? That should be discussed. A comparative discussion will further strengthen the findings.  Several other recent studies have reported the salt and drought stress tolerance by expressing numerous genes like TaNCL2-A, TaGPX1-D etc, where the change in the different parameters have been mentioned like SOD, Cat activities and various others have been mentioned. Besides, numerous studies reported the changes in antioxidants activities like MDHAR, AAO, in addition to GR, GPX, SOD and Cat, during these stress condition. I am surprised how these gene were not identified while interaction analysis, these are the major responsive genes including the calcium signalling genes. In the interaction result, why only WRKY are shown, authors may see the interaction with other genes also.

And at last, authors did not performed any functional study. They should atleast do the characterization of one genes in either homo or heterologous system to get the Ms published in IJMS.

Comments on the Quality of English Language

 Moderate editing of English language required

Reviewer 2 Report

Comments and Suggestions for Authors

The paper makes a complete study of the WRKY in a woody plant. The study is complete, experiments well performed and well presented, and provides plenty of new and interesting information.

I only have three points:

a) Authors compare the genes with arabidopsis, but why? Arabidopsis is an herbaceous plant and a  standard model for plant molecular biology but the  relation to Phoebe is not obvious. To contextualize this I recommend to include a phylogenetic tree with the most relevant plants, crops or tress to determine the evolutionary distances among the two plants compared.   

b) Authors state in the text that there are many descriptions of WRKY genes in different plants. Please include a table with several plants where WRKY gene plants hace been described and how many paralogues are found in each genome. This together with the phylogenetic trees will help to contextualize the study,

c) Which is the n number for figures 8, 9 and 10? Error bars should represent standard error, and not standard deviation, please correct. which statistical method was used to calculate the letters in figure 9. Please include this information in the figure legend.